# Hybrid Nanoparticle/DNAzyme Electrochemical Biosensor for the Detection of Divalent Heavy Metal Ions and Cr^3+^

**DOI:** 10.3390/s23187818

**Published:** 2023-09-12

**Authors:** Evangelos Skotadis, Evangelos Aslanidis, Georgios Tsekenis, Chryssi Panagopoulou, Annita Rapesi, Georgia Tzourmana, Stella Kennou, Spyridon Ladas, Angelos Zeniou, Dimitris Tsoukalas

**Affiliations:** 1Department of Applied Physics, National Technical University of Athens, 15780 Athens, Greece; evaaslani@central.ntua.gr (E.A.); chryssipanagopoulou@mail.ntua.gr (C.P.); rapesisannita@yahoo.com (A.R.); gtzourmana@mail.ntua.gr (G.T.); dtsouk@central.ntua.gr (D.T.); 2Biomedical Research Foundation, Academy of Athens, 4 Soranou Ephessiou Street, 11527 Athens, Greece; gtsekenis@bioacademy.gr; 3Department of Chemical Engineering, University of Patras, 26504 Patras, Greece; kennou@chemeng.upatras.gr (S.K.); ladas@chemeng.upatras.gr (S.L.); 4Institute of Nanoscience and Nanotechnology, NCSR Demokritos, Aghia Paraskevi, 15310 Attiki, Greece; a.zeniou@inn.demokritos.gr

**Keywords:** heavy metal, chromium, biosensor, DNAzyme, nanoparticles

## Abstract

A hybrid noble nanoparticle/DNAzyme electrochemical biosensor is proposed for the detection of Pb^2+^, Cd^2+^, and Cr^3+^. The sensor takes advantage of a well-studied material that is known for its selective interaction with heavy metal ions (i.e., DNAzymes), which is combined with metallic nanoparticles. The double-helix structure of DNAzymes is known to dissociate into smaller fragments in the presence of specific heavy metal ions; this results in a measurable change in device resistance due to the collapse of conductive inter-nanoparticle DNAzyme bridging. The paper discusses the effect of DNAzyme anchoring groups (i.e., thiol and amino functionalization groups) on device performance and reports on the successful detection of all three target ions in concentrations that are well below their maximum permitted levels in tap water. While the use of DNAzymes for the detection of lead in particular and, to some extent, cadmium has been studied extensively, this is one of the few reports on the successful detection of chromium (III) via a sensor incorporating DNAzymes. The sensor showed great potential for its future integration in autonomous and remote sensing systems due to its low power characteristics, simple and cost-effective fabrication, and easy automation and measurement.

## 1. Introduction

Given the current situation regarding the amount and diversity of environmental contaminants, there is an urgent need for efficient and dependable pollutant-detection solutions that can both report and quantify them in natural resources such as air or water. Heavy metal ions (HMIs), pesticides, metalloids, inorganic nutrients, pharmaceuticals, and antibiotics are among the most common analytes, while tap, lake, river, ground, and waste water are the most common environmental matrices. Pollutant detection has, in turn, proven to be a challenging task due to the great variability and complexity of the sample matrix, thus requiring highly sensitive detection methods that enable selective detection with sufficiently low limits [1].

HMIs, which can often be found in water, are one of the main hazardous byproducts of rapid industrialization, and pose one of the main threats to human and animal life alike. Water contamination is extremely critical as the contaminants entering drinking or irrigation water accumulate along the food chain. In addition, HMI contamination can also harm plants and soil; HMIs have severe adverse effects on plant fertility, and in some cases can even result in plant death. Cadmium (Cd^2+^) is one of the most common and dangerous trace metal ions that can permeate in the environment, particularly in the water [2,3]. Cadmium’s half-life is about 10–30 years [4], i.e., it does not naturally degrade, but instead accumulates in vital organs such as the intestine, liver, and kidneys. [5]. Cd^2+^ toxicity can be clearly seen in its interference with many of the necessary nutrients for proper cellular function; this inhibits the absorption of iron, calcium, or zinc in the intestines [6], resulting in slow growth, dysfunction of the liver and kidneys, anemia, and bone damage. Cadmium can also occupy protein-binding sites, interfere with the secretion and regulation of protease-related hormones, reduce the antioxidant capacity of the body, and damage organs and tissues. Lead (Pb^2+^) is also one of the most persistent and toxic pollutants. Lead pollution sources mainly include waste batteries, automobile exhausts, mining, and metal smelting [7,8]. This can cause Pb^2+^ to enter the food chain and pollute water resources [9,10]. Even low doses of Pb^2+^ can accumulate in the human body and are difficult to excrete; long-term exposure or excessive intake of Pb^2+^ can cause serious disease in organs or ecosystems [11]. In particular, Pb^2+^ can be more detrimental to children than adults by impeding brain and height development [12]. Chromium is commonly used in steel processing, coating materials, pigments, and catalysis; it is mutagenic and carcinogenic [13] and its toxicity has raised serious environmental concerns. Chromium has nine different oxidation states, with trivalent Cr^3+^ and hexavalent Cr^6+^ (present as Cr_2_O_7_^2−^ in acid and CrO_4_^2−^ in alkaline solutions) being the most common ones. The toxicity of chromium is strongly related to its oxidation state [14]. For example, a trace amount of Cr^3+^ is needed for glucose and fat metabolism and, thus, Cr^3+^ is considered to be essential for life [15]. However, Cr^3+^ is still toxic at high concentrations and is currently classified as a group 3 carcinogen [16].

Several analytical methods have been developed that can detect heavy metals in water; atomic fluorescence spectroscopy (AFS) [17], atomic absorption spectroscopy (AAS) [18], inductively coupled plasma atomic emission spectroscopy (ICP-AES) [19,20], and inductively coupled plasma-mass spectrometry are some of the most common methods for HMI determination. However, these methods are often time consuming, expensive, require sophisticated equipment and highly trained personnel, and are not practical for in situ detection [21]. Electrochemical methods have shown the potential to be just that—easy, cost effective, and precise methods that, because of their easy automation and integration with read-out circuits and electronics, can eventually be used for on-site target detection. Sensors in general and, most importantly, nanomaterial-based sensors, have the capacity to assist analytical techniques via on-site screening of water bodies and, eventually, replace many of the water-quality sensors that are used today for HMI detection [1]; at the same time, nanosensors offer low-cost and low-power consumption, as well as portability when integrated with point-of-care systems that can also be combined with emerging technologies such as the Internet of Things (IoT) [22]. Nanomaterial-enabled water-quality sensors rely on a number of transduction methods that can be applied for the detection of environmental contaminants and particularly HMIs, such as optical sensors (i.e., colorimetric and fluorescence) for the detection of Cd^2+^, Pb^2+^, Hg^2+^, and Cu^2+^ [23,24,25,26,27,28,29]; electrochemical sensors [30]; and field effect transistors [31]. The use of functional biomaterials such as DNAzymes is also common in the development of highly sensitive and selective biosensors for HMI detection. DNAzymes are enzymatic, single-stranded (ss), synthetic DNA sequences that present a high specificity towards metal ions; binding of metal ions to specific sites in the DNA sequence leads to DNA cleaving and to its eventual breakdown. There are a wide range of DNAzymes that have been reported to enable the bio-recognition of HMIs, such as Pb^2+^, while in the case of Cd^2+^, the development of suitable and selective DNAzymes is more challenging due to the thiophilicity of Cd^2+^ [32]. The main competing technologies for DNAzyme0based sensors are colorimetric or fluorescent sensors [33,34,35] and electrochemical sensors, i.e., amperometric [36,37] or impedimetric sensors [38,39]. Nanomaterials are also often combined with DNAzymes; among the most common scenarios is the use of nanomaterials as optical quenchers [40], as enhancers for the optical properties of respective sensing systems (e.g., the utilization of localized surface plasmon resonance (LSPR) characteristics of Au nanoparticles (NPs) [41], the use of quantum dots so as to achieve chemiluminescence resonance energy transfer (CRET) signals [42], or as appropriate platforms for DNA immobilization [43]. There are many articles and reviews [21,44] dedicated to the use of DNAzymes for Pb^2+^ detection; the detection principle is again either colorimetric, fluorimetric, or electrochemical, while typical limits of detection (LoD) range between 0.5–400 nM, 100 pM–500 nM, and 6.4–910 pM for colorimetric, fluorescence, and electrochemical sensors, respectively. Typical LoDs in the case of Cd^2+^ DNAzyme-based biosensors fall in the range of 1–11.3 nM [32,45]; in the case of a non-specific DNAzyme-based biosensor fabricated using a field-effect transistor (FET) and single-walled carbon nanotubes (SWNTs), the use of a mathematical model and data analysis results in a Cd^2+^ LoD of 34 pM [46]. It is also worth noting that in the case of Cr^3+^, there are very few publications related to DNAzyme-based sensors, focusing either on the use of solid state nanochannels in tandem with DNAzymes in order to modify the ionic current in the presence of Cr^3+^ ions within the nanochannel [47], the development of a molecular beacon for the detection of Cr^3+^ and Cr^6+^ [48], and on fluorescence [49] or colorimetric sensing concepts [50].

In the current paper, an electrochemical biosensor based on platinum (Pt) NPs and DNAzymes is proposed for the simultaneous and label-free detection of HMI targets, namely Pb^2+^, Cd^2+^, and Cr^3+^. The two-dimensional NP film is deposited in between interdigitated electrodes (IDE), serving as “expanded” nano-gapped electrodes. Target-appropriate DNAzymes were chosen for the selective detection of all of the target materials; apart from its catalytic activity in the presence of HMIs, the DNAzymes layer offers enhanced device conductivity, which is reduced in the presence of the target material due to the collapse of conductive DNAzyme inter-nanoparticle bridging. Two functional groups have been employed for the surface immobilization of DNAzymes (i.e., thiol and amino groups). The sensor was able to detect Pb^2+^, Cd^2+^, and Cr^3+^ concentrations that are well below their respective permitted levels in tap water. This group has previously reported on the combination of biomaterials such as DNAzymes, oligonucleotides, and aptamers with metallic NPs [51,52,53] for the development of bio-sensing devices. The current work expands on these results by further optimizing the device concept, resulting in the development of a highly sensitive and flexible sensing platform that is capable of detecting and quantifying three distinctive HMIs. The proposed hybrid nanomaterial/DNAzyme electrochemical biosensor serves as a flexible and modular biosensing platform that can be expanded so as to enable the detection of additional environmental contaminants by accommodating additional DNAzymes. It is also worth noting that this is one of the few reports discussing the successful detection of Cr^3+^ using DNAzymes, regardless of the sensing technique, while to the best of the authors’ knowledge, this is the sole report regarding electrochemical biosensors. In addition, the results discussed herein highlight the importance and impact of DNA functional groups (i.e., amino and thiol) on the biosensor’s performance, allowing for the optimization of the biosensor; optimization is achieved both in terms of improved biosensor performance as well as in terms of simpler and faster fabrication. Overall, the biosensor presented herein poses as a unique sensing solution within the field of electrochemical biosensors as it utilizes the NP layer in a radically different manner than most electrochemical-based sensing systems and by taking advantage of double stranded (ds) DNA’s electrical properties [54]. In addition, the biosensors stand out as an attractive and cost-effective solution by avoiding many of the common drawbacks encountered in most nanomaterial/DNAzyme electrochemical-sensing systems, namely arduous and expensive development that often involves the use and combination of numerous and wildly varying materials and/or complicated fabrication techniques. At the same time, it relies on a relatively simple and low-power characterization method (resistance measurement under a 1V bias), rendering it suitable for future integration to portable and remote environmental-monitoring systems, as well as to water treatment and remediation platforms.

## 2. Materials and Methods

### 2.1. Interdigitated Electrode & Pt Nanoparticle Fabrication

Silicon substrates with a thermal SiO_2_ oxide layer of 300 nm in thickness were used for the development of the biosensor. Gold IDEs with a 10 μm inter-finger distance were patterned on top of the oxidized substrates via optical lithography and an e-gun metallization step. Titanium (10 nm in thickness) was used as an intermediate adhesion layer between SiO_2_ and Au, resulting in a total IDE thickness of 40 nm. Naked Pt NPs were deposited using a physical vapor deposition technique, i.e., DC magnetron sputtering [51,52,53]; sputtering is a room-temperature technique that allows for the simultaneous control of both the nanoparticle size (controlled via the sputtering target to the deposition-substrate distance), as well as nanoparticle surface coverage and density (dependent on the overall deposition time) and, thus, the conductivity and resistance of the device [51,52]. Device resistance was monitored in situ during NP deposition and was interrupted as soon as the desired resistance was achieved. Devices used in the current paper were prepared with a NP surface coverage just below the percolation threshold for optimum device sensitivity and according to previous results reported by this group [51,52].

### 2.2. Materials and Reagents Used for Functionalization

All of the employed reagents were of analytical grade and were obtained from Merck (Merch SA). All of the buffers were prepared with deionized water (18.2 MΩ cm at 25 °C resistivity) from a Millipore MilliQ system. Oligo nucleotides employed for the formation of the ds DNAzymes Gr5, detection of Pb^2+^, Cd^2+^, and Cr^3+^ ions were purchased from Integrated DNA Technologies, BVBA (Leuven, Belgium). Their sequences were Gr5 catalytic strand: GTTCGCCATCTGAAGTAGCGCCGCCGTATAGTGACT; BN-Cd16 catalytic strand: GTTCGCCATCTTCCTTCGATAGTTAAAATAGTGACT; and Ce13d catalytic strand: GTTCGCCATAGGTCAAAGGTGGGTGCGAGTTTTTACTCGTTATAGTGACT.

Following hybridization with a common substrate strand, whose sequence was AG-TCACTATrAGGAAGATGGCGAAC, they were capable of recognizing Pb^2+^, Cd^2+^, and Cr^3+^ ions, respectively, and inducing self-catalysis at the ribonucleotide base. The substrate strand was modified at 5′ with either an amino C6 linker or a thiol C6 linker to allow for its immobilization onto the sensor surfaces. The following materials and reagents were used: materials (3-Aminopropyl)triethoxysilane (APTES) (an aminosilane with alkoxysilane molecules); glutaraldehyde and ethanol; Phosphate Buffered Saline (PBS) of pH = 7.4; phosphate buffer 1 M, pH = 8, 0.001% tween20; ethanolamine; MOPS buffer (3-(Nmorpholino)propanesulfonic acid) (50 mM MOPS/25 mM NaCl, pH = 7.5) and MES buffer (2-(N-morpholino)ethanesulfonic acid) (50 mM MES/25 mM NaCl/0.8 mM Phosphate Buffer, pH = 6); and 6-mercapto-1-hexanol (MCH).

### 2.3. Surface Functionalization and DNAzymes Immobilization

All of the preparation steps were conducted at room temperature, unless stated otherwise. Two different techniques were used for the functionalization of the surfaces. The first method used amino-modified DNA sequences, while in the second method, thiol-modified DNA sequences were employed. It must be emphasized that during every step of the following processes, only the active area of the sensors was processed, namely the IDEs surface area.

For the amino-modified DNA immobilization technique, the following steps were applied, as described in Figure 1. The first step involved the activation of the sample’s surface (SiO_2_) with oxygen plasma; in the second step, the samples were functionalized with APTES. This was essential for the next step of the process, where the covalent bond between glutaraldehyde with the alkoxysilane molecules was established. APTES was used in a 5% *v*/*v* solution of ethanol and DI water (solution 1). The chips were kept in dark room conditions for 2 h and were then rinsed with a DI water and ethanol solution (5% *v*/*v*) and baked at 110 °C for 1 h. Glutaraldehyde was dissolved in a 5% PBS 1× buffer solution; the samples remained in this solution for 1 h and were subsequently washed with the same buffer and DI water, and finally gently blow dried with N_2_. The surface functionalization step was followed by the probe immobilization and target hybridization steps, respectively. Initially, the ss DNA substrate probes (aminated oligonucleotides) were immobilized on the second aldehyde group of glutaraldehyde. DNA deposition was performed via drop-casting from 10 μΜ phosphate buffer. After covalent coupling, ethanolamine was used in a 10 μM solution in phosphate buffer to remove any non-specifically bound catalytic strands from the surface and to act as an interaction barrier between the immobilized single DNA strands. The final step of the process involved the hybridization of the DNAzyme sequences with the immobilized substrate strands, which was performed from either a 5 μM MOPS buffer in the case of Pb^2+^ and Cd^2+^ detection, or a 5 μM MES buffer in the case of Cr^3+^ detection. It should be noted that after every step, the samples were washed with the same buffer solutions the drop-casted substance was dissolved in, and the rinsed with DI water and blow-dried with N_2_, while in humid conditions.

For the thiol-modified DNA immobilization technique, there was no need for surface modification prior to probe immobilization and target hybridization (Figure 2). A 5 μΜ phosphate buffer with the dissolved ssDNA substrate probes was heated at 95 °C for 5 min, and then deposited on the IDEs via drop casting. After 1 h in humid conditions, MCH was employed in the same phosphate buffer solution, in order to convey an identical blocking effect as ethanolamine on the amino-modified DNA sequences. Finally, the DNAzyme sequences were hybridized with the immobilized substrate strands using the same MOPS or MES buffer solutions as in the amino-modified DNA immobilization technique.

After the successful formation of the ds DNAzymes film on the biosensor’s surface, the device was blow dried using gentle N_2_ flow and could be immediately used for heavy metal ion detection and recognition while exposed to room temperature and humidity and without any special requirements. The device could be also stored in humid conditions and in a temperature between 4 and 5 °C; the proposed biosensors could successfully detect all three heavy metal ion targets, even if stored in such conditions for over a month.

### 2.4. Surface Characterization and XPS Analysis

Scanning electron microscopy (SEM) analysis was conducted for sensors functionalized with both amino and thiol linkers and for every step of the immobilization process; SEM characterization showed that electron charging on the sensors’ surface increased with each additional layer, which prevented detailed SEM imaging. In addition, each of the DNAzymes functionalization steps (as described in Figure 1 and Figure 2) was verified using fluorescence microscopy. To that end, all of the DNA strands used in this work (substrate and enzymatic/catalytic) were labeled with fluorescent tags, as discussed in [51].

In order to confirm the immobilization and loading of the DNAzymes layer on the sensor’s surface, as well as to observe any possible differentiation between amino and thiol linkers, X-ray photoelectron spectroscopy (XPS) analysis was employed. XPS analysis of plain Si/SiO_2_ silicon slabs, Si/SiO_2_ slabs with Pt NPs, Si/SiO_2_ with thiol or amino-Dnazymes, and Si/SiO_2_ with Pt NPs and amino or thiol-DNAzymes was conducted using a MAX200 system. The XPS measurements took place in the analysis chamber of MAX200, at room temperature and ~4 × 10^−8^ mbar pressure, using conventional non-monochromatic MgKα X-rays and a Hemispherical Electron Energy Analyzer (SPECS EA200) with Multi-Channel Detection properly calibrated according to ISO15472 and ISO24237. The analyzer operated under conditions optimized for a better signal intensity (constant pass energy of 100 eV, maximum lens aperture). The measurements always took place along the specimen surface normal (0° take-off angle). The analyzed area was laterally defined by a ~4 × 7 mm^2^ rectangle, centrally placed over each specimen, so as to always take measurements within the specimen surface. The maximum analyzed depth, from the outermost surface inwards, was 15 nm, with the signal intensity decreasing roughly exponentially with increasing depth.

XPS analysis confirmed that without the presence of the Pt NP layer (i.e., for bare SiO_2_ substrates), there was no organic loading in the case of thiol-modified DNAzymes (distinct XPS peaks for N, C, and P) on the biosensor’s surface. On the contrary, amino-modified DNAzymes could successfully attach on both bare and Pt NP-modified surfaces. Quantitative analysis of the XPS results confirmed these observations, indicating that the number of deposited biomolecules was considerably less for thiol-modified DNAzymes compared with the amino-modified DNAzymes, which, again, is to be expected as the functionalized bio-molecules could attach on the entire surface of the biosensors.

### 2.5. Sensor Characterization

The finalized biosensors were characterized in a homemade electrochemical cell using a Keithley 2400 for resistance monitoring under a 1 V DC bias; the actual characterization process has been described in length in [51]. In Figure 3, the typical dynamic sensor-response can be seen for a sensor measured as described in [51]. The introduction of a buffer solution on the top of the sensor resulted in a drastic decrease in sensor resistance, while a new steady-state for sensor resistance was achieved in fewer than 10 s. The introduction of a buffer solution spiked with the given HMI concentration was detected as an increase in the sensor’s resistance.

## 3. Results and Discussion

The charge-transport mechanism for NP films developed via sputtering and with a surface coverage just below the percolation threshold has been discussed in length in previous publications by this group [51,52]. In short, the charge-transport mechanism in this case is governed by quantum mechanical phenomena such as tunneling and/or variable range hopping, etc.; devices that fall in this regime are often associated with a thermally activated Arrhenius-type mechanism for conductivity that resembles that of a semi-conductor [51]. It is thus obvious that such devices are extremely sensitive to any variation in their environment, such as any decrease or increase in inter-nanoparticle distance, changes in the dielectric constant of the surrounding medium, and the introduction of functionalization materials between distinct NPs or between NP aggregates. On the other hand, dsDNA exhibits interesting electrical properties and is known to facilitate charge transport via tunneling over shorter oligonucleotides or via multi-step hopping over longer DNA paths [54]; this is also the case for ssDNA, albeit with conductivity that is order of magnitudes lower than that of dsDNA [55]. The electrical properties of dsDNA and its respective charge transport, using the HOMO/LUMO and the π-electronic system of stacked base pairs, have been studied extensively, yet many phenomena still need to be understood; more specifically, distinctive DNA base pairs have been found to promote charge transport, while others act as electric barriers [54]. Overall, dsDNA conductivity in aqueous environments relies both on base pair π-stacking as well as charge transport via the outer sphere of the sugar phosphodiester backbone of the DNA, including bound water molecules. As discussed in previous publications by this group, the stacking of conductive molecular bridges such as DNAzymes between the NP film leads to enhanced conductivity [51]; in that case and as a proof of concept, the NP film was functionalized with one specific DNAzyme that showed a high affinity towards Pb^2+^. As expected, the conductive bridging offered by the DNAzyme collapsed selectively in the presence of Pb^2+^ ions, translating to an increase in the as-measured resistance of the device.

Herein, our proposed multi-sensing platform was expanded towards the detection of Pb^2+^, Cd^2+^, and Cr^3+^ by using additional and highly specific DNAzymes; it is worth noting that despite the numerous reports in the literature regarding fluorimetric and colorimetric sensors for Cr^3+^ detection [56], or reports discussing the development of FET sensors [57], up until now, there has been a very limited number of publications related to the detection of Cr^3+^ ions using DNAzymes, regardless of the detection principle. In addition, and in order to investigate the role of different functionalization modifications, our ds catalytic strands were functionalized with either thiol or amino functional groups. In Figure 4, the response of the proposed biosensors towards Pb^2+^, Cd^2+^, and Cr^3+^ for thiol-modified catalytic strands can be seen. The results correspond to a relative change in resistance (ΔR/R_0_%); as a result of the cleavage of the substrate strand and its dissociation into two smaller fragments, there was an increase in the measured resistance of the device [51]. The mean base resistance or R_b_ (R_b_: initial device resistance) of the sensors used in these experiments was in the range of 500–950 kΩ, while it had a standard deviation of 5.6%. For the detection of each distinctive HMI concentration, 10 different DNAzyme biosensors were used in total in order to calibrate the biosensors. The standard deviation of these precision measurements was in the range of 0.28% and 1.2%. The sensors had a limit of detection (LoD) of 0.8 nM, 1 nm, and 10 nm for Pb^2+^, Cd^2+^, and Cr^3+^ respectively. The response time of the sensors was between 7 and 19 s (response time is the time needed for the sensor to reach 70% of its steady state for a specific HMI concentration). The cross-sensitivity and selectivity of the biosensors was also examined by introducing a buffer solution containing an HMI that was non-specific to the respective DNAzyme; the introduction of the non-specific HMI was performed before the introduction of the target-specific HMI.

Figure 5 shows the response of the hybrid biosensors towards Pb^2+^, Cd^2+^, and Cr^3+^ for amino-modified catalytic strands. Again, the R_b_ of the biosensors was chosen to be in the range of 500–950 kΩ. As in the case of thiol-modified DNAzymes, 10 distinctive DNAzyme biosensors were used for calibrating the sensor response towards each HMI concentration. The standard deviation of the precision measurements was between 0.3% and 0.9%. In this case, the sensors were characterized by a LoD of 20 nM, 20 nm, and 44 nm for Pb^2+^, Cd^2+^, and Cr^3+^, respectively, and a response time between 6 and 20 s. Finally, their cross-sensitivity/selectivity was evaluated towards an HMI that was non-specific to the respective DNAzyme.

As can be seen in Figure 6, the DNAzyme modification had an effect on device performance and sensitivity. To be more specific, thiol-modified DNAzymes showed greater sensitivity when compared with their amino-modified counterparts. Platinum NPs have found wide use as promising nanomaterials for biomedical applications [58,59]; this would not be possible if not for their facile bio-conjugation with a wide range of bio-materials. There are many reports on the successful immobilization of both thiol [60] and amino [61]-modified biomolecules (directly on the surface of platinum nanoparticles, i.e., without the necessity for an intermediate functionalization layer such as APTES and GOPTS), intercalating aptamers, oligos, dsDNA, etc., in-between Pt NPs. Herein, we followed this approach for both thiol and amino modified DNAzymes.

The schematic in Figure 7 showcases the DNAzyme distribution on top of two-dimensional Pt NP films, as produced via the sputtering technique; the schematic aims to provide a realistic depiction of the device, in accordance with TEM characterization results [51,52,53]. As can be seen, there was a random distribution of inter-nanoparticle gaps (noted as “d”) that could be under 1 nm (where charge transport via tunneling can occur) or well over 2 nm (where charge transport could only occur via the hybridized DNAzymes-layer, and for appropriate distances that matched the length of the dsDNA [51]). The dissociation of the substrate strand after the successful recognition of target HMIs had a significant effect on conductivity; to be more specific, the collapse of conductive DNA bridging resulted in an increase in device resistance for inter-nanoparticle gaps over 2 nm [51]. For thiol-modified DNAzymes, it is clear that the sensing mechanism was dominated by conductive DNA bridging. In the case of amino DNAzymes without any functionalization on the sensor’s surface, our experiments did not result in the detection of any HMIs in sufficiently low concentrations (i.e., under 200 nM); hence, they were not included in this paper. At this point, it is worth noting that the quality of the anchoring group (i.e., amino or thiol) can play a crucial role in device conductivity with thiol modification groups, offering improved charge transport due to stronger bonds with the metallic NPs [54]. In our case, the low sensitivity observed in the case of amino DNAzymes led to the incorporation of a surface functionalization process (as described in the experimental section of the paper) in order to enhance the attachment of the amino groups on the Pt NP film and, hence, improve the device performance. The incorporation of surface functionalization layers, such as the aminosilane APTES followed by its covalent bonding to glutaraldehyde and ultimately to ssDNA strands through carbonyl groups, may have enhanced the amino to nanoparticles bonding strength, but at the same time it seemed to affect and disturb charge transport by forcing it to transit through additional functionalization layers and inter-layer chemical bonds. In addition, and as can be seen in Figure 7, the immobilization of DNAzymes on the SiO_2_ substrate physically obstructed the inter-nanoparticle conductive bridging, thus resulting in a lower sensitivity. In conclusion, apart from the improved sensing response in the case of thiol-modified DNAzymes, thiol-based devices offer a much simpler fabrication and significantly shorter preparation time and are hence extremely cost-effective when compared with their amino DNAzyme counterparts. It is worth noting that the proposed devices could be easily used in tandem with advanced functional materials intended for water treatment (e.g., carbon-based materials) [62,63,64] in order to offer a holistic solution for water monitoring and remediation. Finally, the sensors cοuld be reused, as discussed in [51]; however, the regeneration process has been proved to be cumbersome and unnecessary. That is because a small fragment of the substrate strand remained attached to the surface. Regeneration of the surface would require the complete removal of this fragment through the disruption of the covalent bond tethering to the sensor surface. This, in turn, would necessitate the functionalization of the sensor surface again, which is a costly and inefficient process.

## 4. Conclusions

A hybrid electrochemical biosensor for the detection of three distinctive metal ions (i.e., Pb^2+^, Cd^2+^, and Cr^3+^) has been presented. This is one of the few DNAzyme-related works that discusses the successful detection of Cr^3+^ ions using sensors, regardless of the choice of detection principle, and the sole report on successful electrochemical detection. The sensor is based on the combination of noble metallic nanoparticles (i.e., platinum) and DNAzymes. The biosensor was functionalized with target-specific catalytic DNA strands that can bind on the sensor’s surface via two different anchoring groups that are often used in the literature, namely thiol and amino functional groups, in order to study their effect on sensor performance. The biosensor was ultimately able to detect all three heavy metal ions in sufficiently low concentrations that are well below their permitted levels in tap water, for both anchoring groups. However, our results show that the direct attachment of catalytic DNAs on the noble nanoparticle surface (i.e., in the case of thiol-modified DNAzymes), results in sensors with an improved performance and much simpler and faster fabrication. A surface-functionalization technique has been proposed in order to enhance the binding strength of amino-modified DNAzymes to the surface, which resulted in significant improvement in sensor performance; still, sensors utilizing thiol-modified DNAzymes continued to outperform their amino counterparts.

The sensors have been proven to be reliable with a good sensitivity, precision, and sufficient dynamic range that can be further expanded towards increased concentrations if needed. In addition, and as indicated by previous results [51], the sensors are re-usable, but do require additional re-hybridization steps after target detection between catalytic strands and substrate strands. At the same time, the biosensors offer simple instrumentation and measurement, i.e., device resistance under a low bias (1 V). This further highlights the simplicity and added value of the device, while its low-power properties along with its easy automation render it well suited for remote and autonomous environmental monitoring systems or water-treatment systems, going forward into the IoT era. It is also worth noting that the proposed biosensor can be developed as a multi-sensing array for the simultaneous detection and screening of additional HMIs, while the range of candidate environmental contaminants can be expanded. The eventual goal of the present work is the integration of the biosensors in a single, disposable, and low-cost platform, which will allow for the multiplexed detection of all three HMIs in a single measurement. Future work entails revisiting the biomaterial protocols so as to further facilitate the reusability of the device, as well as revisiting the direct attachment of amino-modified catalytic strands on the nanoparticle’s surface in order to investigate any underlying phenomena that contribute to the sensor’s sensitivity.

## Figures and Tables

**Figure 1 sensors-23-07818-f001:**
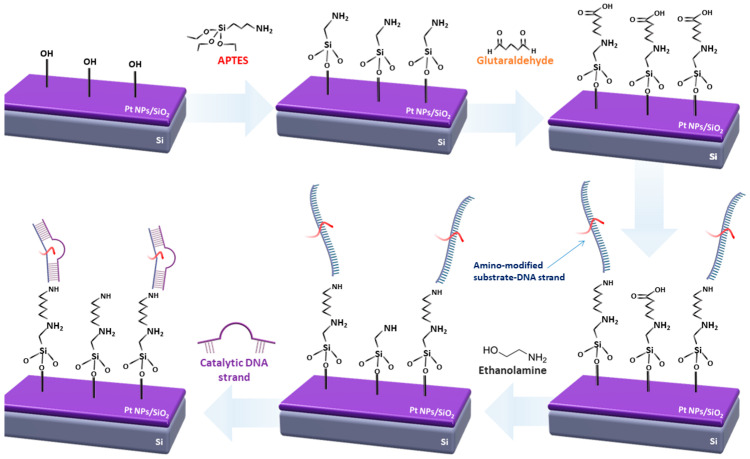
Schematic representation of the immobilization process for amino-modified DNAzymes.

**Figure 2 sensors-23-07818-f002:**
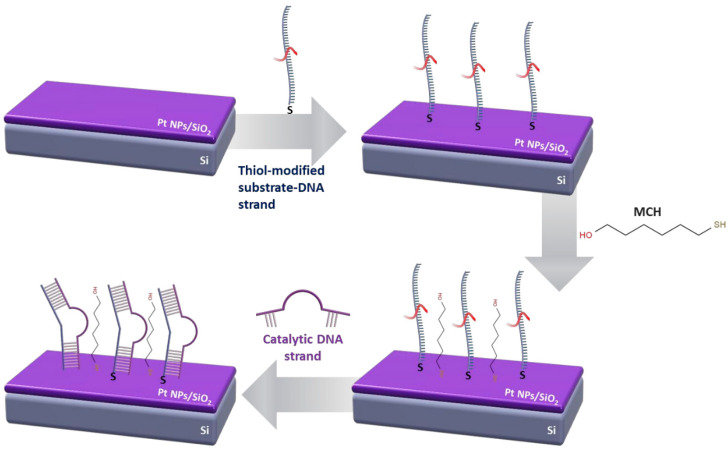
Schematic representation of the immobilization process for thiol-modified DNAzymes.

**Figure 3 sensors-23-07818-f003:**
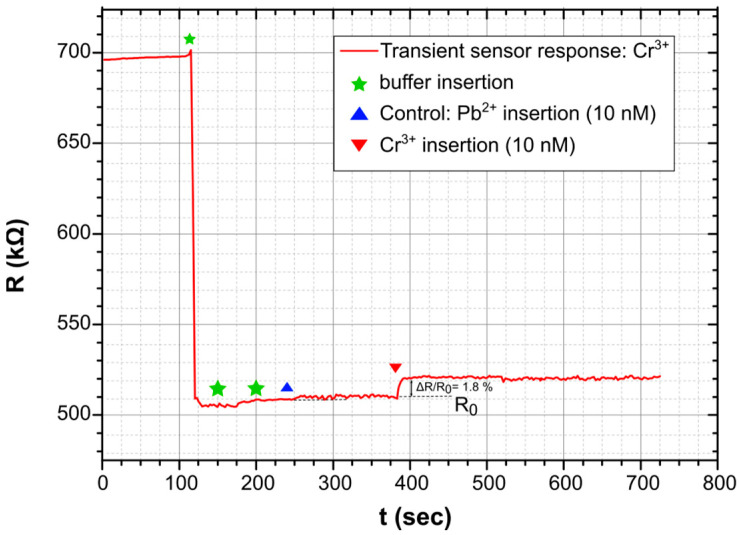
Typical dynamic response of a biosensor when combining Pt nanoparticles and thiol modified DNAzymes. Detection of 10 nM of Cr^3+^. In the current study, sensors that incorporate thiol-modified DNAzymes for lead and cadmium, as well as sensors that incorporate amino-modified DNAzymes for all heavy metal ions, feature a similar dynamic response.

**Figure 4 sensors-23-07818-f004:**
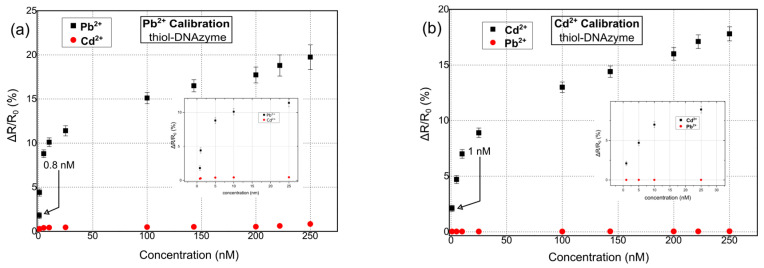
Relative resistance response % for hybrid nanoparticle/DNAzyme biosensors, incorporating a thiol anchoring group. Sensor response is calculated as ΔR/R_0_, where R_0_ is the resistance of the biosensor after the introduction of the buffer solution and prior to the introduction of any heavy-metal ion solution. (**a**) Sensor calibration for varying concentrations of Pb^2+^. (**b**) Sensor calibration for varying concentrations of Cd^2+^. (**c**) Sensor calibration for varying concentrations of Cr^3+^. (**d**) Cumulative results for the detection of all three heavy metal ions: Pb^2+^, Cd^2+^, and Cr^3+^.

**Figure 5 sensors-23-07818-f005:**
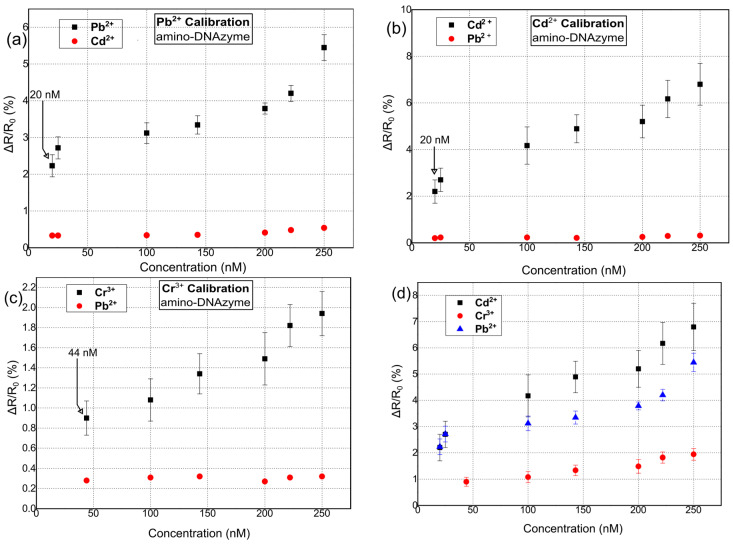
Relative resistance response % for hybrid nanoparticle/DNAzyme biosensors, incorporating an amino-anchoring group. Sensor response is calculated as ΔR/R_0_, where R_0_ is the resistance of the biosensor after the introduction of the buffer solution and prior to the introduction of any heavy-metal ion solution. (**a**) Sensor calibration for varying concentrations of Pb^2+^. (**b**) Sensor calibration for varying concentrations of Cd^2+^. (**c**) Sensor calibration for varying concentrations of Cr^3+^. (**d**) Cumulative results for the detection of all three heavy metal ions: Pb^2+^, Cd^2+^, and Cr^3+^.

**Figure 6 sensors-23-07818-f006:**
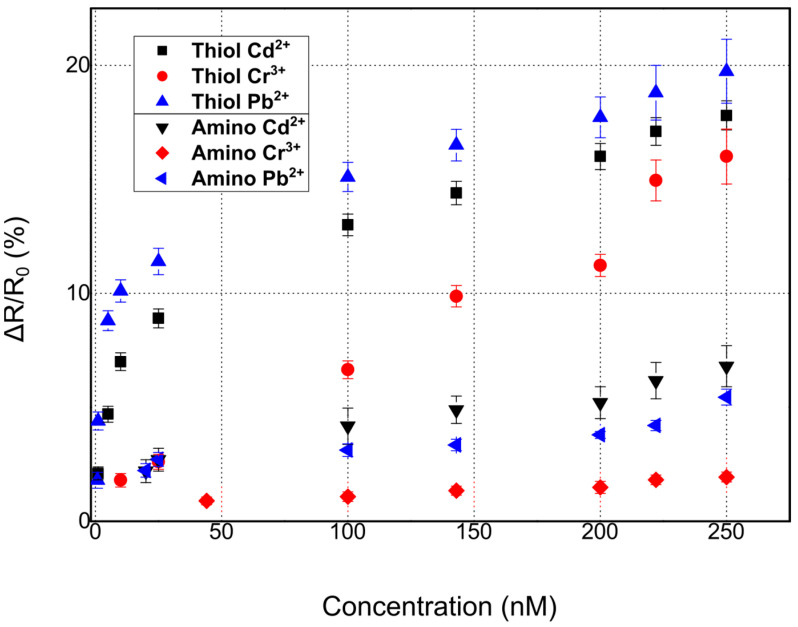
Comparative chart for the detection of all three heavy metal ions, Pb^2+^, Cd^2+^, and Cr^3+^, using two distinctive anchoring groups: thiol and amino-modified DNAzymes.

**Figure 7 sensors-23-07818-f007:**
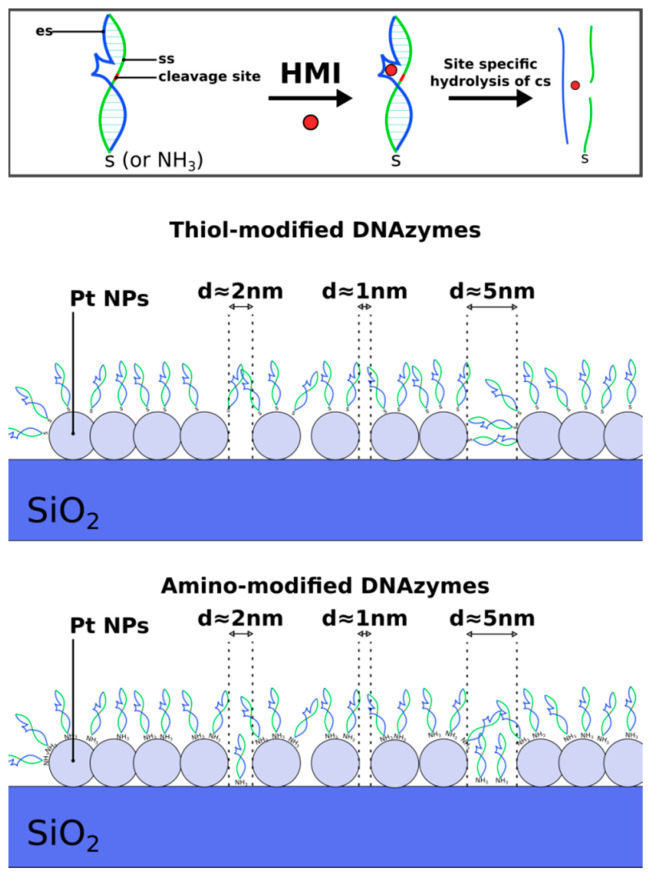
Schematic representation of the DNAzyme distribution on top of the two-dimensional platinum (Pt) nanoparticle (NP) film, for amino and thiol-modified DNAzymes. The enzyme strand (es), substrate strand (ss), and cleavage site can be seen in the respective DNAzymes schematic; upon recognition of a heavy metal ion (HMI) target, the substrate strand is cleaved. The Pt NP film offers a wide range of inter-nanoparticle gaps (noted as “d”) that can both be under 1 nm and well over 2 nm.

## Data Availability

Data supporting reported results can be found here: https://figshare.com/articles/dataset/Hybrid_nanoparticle-DNAzyme_electrochemical_biosensor_for_the_detection_of_divalent_heavy_metal_ions_and_Cr3_/23790675, accessed on 1 August 2023.

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
