# Peer review of "Hybrid Nanoparticle/DNAzyme Electrochemical Biosensor for the Detection of Divalent Heavy Metal Ions and Cr3+"

_sensors, 2023, doi:10.3390/s23187818_

Round 1

Reviewer 1 Report

In this study, the authors developed and introduced an electrochemical heavy metal ion sensor using Pt nanoparticles and DNAzyme immobilized IDEs. They explained the experimental results and related theories well. I believe that this paper will be helpful to readers interested in electrochemical sensor-related research. Before accepting this manuscript, I advise the authors to add the following experiments and explanations.

1. In Figure 1 and 2, how to confirm each immobilizing step? Also, how does the resistance response change with each step of immobilization? You can use this result to further explain the differences between the two immobilization processes.

2. Figure 6 is just a combination of Figure 4d and Figure 5d, right? It's not a good idea to include the same data multiple times.

Author Response

Dear editor,

We would like to thank the reviewers for their insightful and constructive comments. In the text below we address the comments raised by Reviewer 1; our changes in the paper are marked in red color.

Reviewer 1

  1. In Figure 1 and 2, how to confirm each immobilizing step? Also, how does the resistance response change with each step of immobilization? You can use this result to further explain the differences between the two immobilization processes.

Each of the immobilization steps (as described in Figures 1 and 2) have been verified using fluorescence microscopy. To that end, all of the DNA strands used in this work (substrate and enzymatic/catalytic) have been labeled with fluorescent tags as discussed in [52]. It is also worth noting that XPS measurements have confirmed the successful immobilization of the DNAzymes, as described in the revised version of our manuscript.

Changes can be found in subsection “2.4 Surface characterization and XPS analysis”, of the experimental section; lines 4-7 refer to the use of fluorescence microscopy.

The resistance of the proposed biosensors has been measured after the substrate-strand and catalytic-strand immobilization steps, in ambient conditions as well as in the presence of a buffer solution. In the case of ambient conditions, an increase in resistance has been observed after the addition of each DNA-strand while no significant differences have been observed between amino and thiol linkers. In the case of resistance measurements while in the presence of a buffer solution, the resistance of the device decreased dramatically for both thiol and amino linkers alike; however, this should be attributed to the presence of the buffer solution. It is worth noting that in order to obtain optimal spatial arrangement of the double stranded DNAzymes the respective film should be in a buffer-solution environment; hence resistance measurements in ambient conditions offer limited insight on the organization of the DNAzymes layer on the sensor’s surface. Unfortunately resistance measurements in-between the various immobilization steps as well as after the completion of the immobilization process, did not provide any additional insight on any potential difference between amino and thiol DNA linkers. 

  1. Figure 6 is just a combination of Figure 4d and Figure 5d, right? It's not a good idea to include the same data multiple times.

This is correct, Figure 6 combines in the same graph experimental data for thiol and amino modified DNAzymes that are previously shown in separate figures. We would like to leave this additional figure since the comparison between thiol and amino DNAzymes is amongst the most significant findings/reports of this paper and it should help in further highlighting the effect of DNAzyme modification in sensor performance. Graphical changes have been made in figure 6 so as to better illustrate our results

Reviewer 2 Report

In this paper, the authors propose a noble metal nanoparticle /DNAzyme composite electrochemical biosensor for the detection of pb2+, cd2+ and cr3+. The topic of this manuscript is interesting. However, major revisions are required and the comments are given below.

1.     Authors should check the manuscript for grammatical errors.

2.     There are too many keywords, just choose 4-5.

3.     Some of the references cited in the introduction section are quite old. Please add the latest literature, e.g. Journal of Bioresources and Bioproducts 2022, 7 (2), 109-115; Journal of Bioresources and Bioproducts 2021, 6 (4), 292-322; Nanoscale 2022, 14, 8216.

4.     Provide a separate introduction to the reagents used in the article in the second part.

5.     In Figure 4, the small images in each image are blurry. If necessary, please create a separate image.

6.     Please unify the positions of the symbols in the pictures.

7.             Please pay attention to the writing of units.

8.     Pay attention to the writing of corner markers.

9.     Has Scanning electron microscope (SEM) analysis and energy dispersive spectroscopy (EDS) analysis been used to evaluate the process of precipitation loading on the biosensor surface? If not, please supplement.

10.   The authors need to provide more discussion on the innovation and significance of this work.

11.   After the sensor adsorbs, how to handle the DNA attached to the surface of noble nanoparticles?

12.   Can this sensor be reused? What about the cycle life?

13.   What are the innovative points of this study compared to similar studies? Are there any performance advantages compare to other studies?

Minor editing of English language is required.

Author Response

Dear editor,

We would like to thank the reviewers for their insightful and constructive comments. In the text below we address the comments raised by Reviewer 2; our changes in the paper are marked in red color.

Reviewer 2

  1. Authors should check the manuscript for grammatical errors.

Our revised manuscript has been rechecked for any spelling or grammatical errors.

  1. There are too many keywords, just choose 4-5.

Keywords have been now reduced to 5.

  1. Some of the references cited in the introduction section are quite old. Please add the latest literature, e.g. Journal of Bioresources and Bioproducts 2022, 7 (2), 109-115; Journal of Bioresources and Bioproducts 2021, 6 (4), 292-322; Nanoscale 2022, 14, 8216.

The respective publications have now been added in our revised manuscript in a discussion regarding water treatment/remediation: Section 3, paragraph 5, lines 31-35.

  1. Provide a separate introduction to the reagents used in the article in the second part.

A separate paragraph has been provided in the experimental section of the article, regarding reagents and materials used: “2.2 Materials and Reagents used for functionalization”.

  1. In Figure 4, the small images in each image are blurry. If necessary, please create a separate image.

Separate high-resolution pdf files have been provided along with our revised manuscript; scalable pdf images will be used for the final version of the article solving any issues with image quality. Unfortunately, the docx version of our revised manuscript uses lower resolution BMP files so as to obtain a low file size.

  1. Please unify the positions of the symbols in the pictures.

Changes have been made in the symbols of Figures 4, 5 and 6 in terms of both position and color, in order to obtain better uniformity.

  1. Please pay attention to the writing of units.

Our manuscript has been proofread for any possible errors regarding the writing of units.

  1. Pay attention to the writing of corner markers.

Numbering has been added in the corner of all Figures; in addition, corner markers have been corrected.

  1. Has Scanning electron microscope (SEM) analysis and energy dispersive spectroscopy (EDS) analysis been used to evaluate the process of precipitation loading on the biosensor surface? If not, please supplement.

Scanning electron microscopy (SEM) analysis has been conducted for sensors functionalized with both amino and thiol linkers and for every step of the immobilization process; SEM characterization showed that electron charging on the sensors’ surface increased with each additional layer, which prevented detailed SEM imaging. 

In order to study the immobilization/loading of the DNAzymes layer on the sensor’s surface as well as to observe any possible differentiation between amino and thiol linkers, the authors have opted for the X-ray photoelectron spectroscopy (XPS) analysis of the biosensors. XPS analysis of plain Si/SiO2 silicon slabs, Si/SiO2 slabs with Pt NPs, Si/SiO2 with thiol or amino-Dnazymes and Si/SiO2 with Pt NPs and amino or thiol-DNAzymes, has been conducted using a MAX200 system. The XPS measurements took place in the analysis chamber of MAX200, at room temperature and ~4x10-8 mbar pressure, using conventional non-monochromatic MgKα X-rays and a Hemispherical Electron Energy Analyser (SPECS EA200) with Multi-Channel Detection properly calibrated according to ISO15472 and ISO24237. The analyser operated under conditions optimized for better signal intensity (constant pass energy of 100 eV, maximum lens aperture).  The measurements always took place along the specimen surface normal (0° take-off angle). The analyzed area was laterally defined by a ~4x7 mm2 rectangle, centrally placed over each specimen, so as to always measure within the specimen surface. The maximum analyzed depth, from the outermost surface inwards, is 15 nm with signal intensity decreasing roughly exponentially with increasing depth. 

XPS analysis confirmed that without the presence of the Pt NP layer (i.e. for bare SiO2 substrates) there is no organic loading in the case of thiol-modified DNAzymes (distinct XPS peaks for N, C and P) on the biosensor’s surface. On the contrary, amino-modified DNAzymes can successfully attach on both bare as well as Pt NP modified surfaces. Quantitative analysis of the XPS results has confirmed these observations, indicating that the amount of deposited biomolecules is considerably less for thiol-modified DNAzymes if compared to amino-modified DNAzymes, which again is to be expected since the functionalized bio-molecules can attach on the entire surface of the biosensors.

Changes regarding XPS results can be found in subsection “2.4 Surface characterization and XPS analysis”, of the experimental section.

  1. The authors need to provide more discussion on the innovation and significance of this work.

The innovative aspects of this work as well as its significance have been discussed in our reply to the reviewer’s final comment.

  1. After the sensor adsorbs, how to handle the DNA attached to the surface of noble nanoparticles?

After the successful formation of the double-stranded DNAzymes-film on the biosensor’s surface (immobilization of substrate strand and hybridization with enzymatic strand), the device is blow-dried using gentle N2 flow and can be immediately used for heavy metal ion detection/recognition while exposed in room temperature/humidity and without any special requirements. The device can be also stored in humid conditions and in a temperature between 4 and 5 °C; our proposed biosensors can successfully detect all three heavy metal ion targets even if stored in these conditions for over a month. In addition, ethanolamine and MCH are used in the case of amino and thiol modified DNAzymes respectively, so as to convey a blocking effect and to remove any non-specifically bound catalytic strands from the surface; furthermore, ethanolamine and MCH act as an interaction barrier between the immobilized single DNA strands.

Changes can be found in subsection 2.3, paragraph 4.

  1. Can this sensor be reused? What about the cycle life?

The sensors cοuld be reused as discussed in [52]; however, the regeneration process has proved to be cumbersome and unnecessary. That’s because a small fragment of the substrate strand remains attached to the surface. Regeneration of the surface would require the complete removal of this fragment through the disruption of the covalent bond tethering to the sensor surface. This, in turn, would necessitate the functionalization of the sensor surface anew, a costly and inefficient process. The goal of the present work is the eventual integration of the biosensors in a single, disposable and low-cost platform, which will allow the multiplexed detection of all three HMIs in a single measurement. 

 Changes can be found in Section 3, paragraph 5, lines 35-43.

  1. What are the innovative points of this study compared to similar studies? Are there any performance advantages compare to other studies?

The biosensors discussed in the current paper present many innovative points compared to the competition. To begin with, the biosensors take advantage of double stranded DNA’s charge transport properties and utilize them in a radically different manner compared to the vast majority of biosensors (mostly optical and fluorescence). In addition, this is the first study comparing between thiol and amino linkers for the functionalization of DNA molecules and how they can affect sensor performance. Finally, this is one of the few papers reporting on the successful detection of Cr(III) ions using target-appropriate DNAzymes regardless of the detection principle while at the same time this is the first electrochemical biosensor report on the detection of Cr(III); the successful use of the Ce13d enzymatic strand for the detection of Cr(III) is of significant added value for the community as it further supports and validates its potential for designing and developing varying biosensing systems for Cr(III) detection.

Regarding the performance advantages of our biosensors, it is worth noting that our proposed biosensor serves as a flexible and modular biosensing-platform that can be expanded so as to detect varying environmental contaminants by accommodating additional DNAzymes. In addition, the biosensors avoid many of the common drawbacks encountered in most nanomaterial/DNAzyme electrochemical-sensing systems, namely arduous and expensive development that often involves the use and combination of numerous and wildly varying materials and/or complicated fabrication techniques. At the same time it relies on a relatively simple and low-power characterization method (Resistance measurement under a 1V bias), rendering it suitable for future integration to portable and remote environmental-monitoring and water-treatment systems. 

The last part of the introduction section as well as the conclusions part of the paper have been changed in order to better reflect the innovative points of our study as well as the biosensor’s advantages over its competition.

Round 2

Reviewer 2 Report

The manuscript has been well revised and could be accepted now.